# Understanding the Function and Mechanism of Zebrafish Tmem39b in Regulating Cold Resistance

**DOI:** 10.3390/ijms231911442

**Published:** 2022-09-28

**Authors:** Renyan Liu, Yong Long, Ran Liu, Guili Song, Qing Li, Huawei Yan, Zongbin Cui

**Affiliations:** 1State Key Laboratory of Freshwater Ecology and Biotechnology, Institute of Hydrobiology, Chinese Academy of Sciences, Wuhan 430072, China; 2Guangdong Provincial Key Laboratory of Microbial Culture Collection and Application, State Key Laboratory of Applied Microbiology Southern China, Institute of Microbiology, Guangdong Academy of Sciences, Guangzhou 510070, China; 3College of Advanced Agricultural Sciences, University of Chinese Academy of Sciences, Beijing 100049, China; 4The Innovative Academy of Seed Design, Chinese Academy of Sciences, Beijing 100101, China; 5College of Fisheries and Life Science, Dalian Ocean University, Dalian 116023, China

**Keywords:** cold resistance, zebrafish, *tmem39b*, tissue damage, immune response, damage repair

## Abstract

Autophagy and endoplasmic reticulum (ER) stress response are among the key pathways regulating cold resistance of fish through eliminating damaged cellular components and facilitating the restoration of cell homeostasis upon exposure to acute cold stress. The transmembrane protein 39A (TMEM39A) was reported to regulate both autophagy and ER stress response, but its vertebrate-specific paralog, the transmembrane protein 39B (TMEM39B), has not been characterized. In the current study, we generate *tmem39b*-knockout zebrafish lines and characterize their survival ability under acute cold stress. We observed that the dysfunction of Tmem39b remarkably decreased the cold resilience of both the larval and adult zebrafish. Gene transcription in the larvae exposed to cold stress and rewarming were characterized by RNA sequencing (RNA-seq) to explore the mechanisms underlying functions of Tmem39b in regulating cold resistance. The results indicate that the deficiency of Tmem39b attenuates the up-regulation of both cold- and rewarming-induced genes. The cold-induced transcription factor genes *bif1.2*, *fosab*, and *egr1*, and the rewarming-activated immune genes *c3a.3*, *il11a*, and *sting1* are the representatives influenced by Tmem39b dysfunction. However, the loss of *tmem39b* has little effect on the transcription of the ER stress response- and autophagy-related genes. The measurements of the phosphorylated H2A histone family member X (at Ser 139, abbreviated as γH2AX) demonstrate that zebrafish Tmem39b protects the cells against DNA damage caused by exposure to the cold-warming stress and facilitates tissue damage repair during the recovery phase. The gene modules underlying the functions of Tmem39b in zebrafish are highly enriched in biological processes associated with immune response. The dysfunction of Tmem39b also attenuates the up-regulation of tissue C-reactive protein (CRP) content upon rewarming. Together, our data shed new light on the function and mechanism of Tmem39b in regulating the cold resistance of fish.

## 1. Introduction

Temperature is a key environmental factor that determines the life activities of fish [1,2]. Abrupt temperature changes caused by extreme weather pose a considerable challenge to both cultured and wild-fish populations [3,4]. Temperature approaching or beyond the lower endurance limit of fishes can severely disturb their physiological, biochemical, and metabolic functions, as well as behaviors [5,6]. Cold stress at the molecular level can affect gene transcription and protein translation rates [7,8,9], and cause oxidative damage to DNA and proteins through inducing the production of reactive oxygen species (ROS) [10,11,12]. A causal relationship was established between the cold and rewarming-induced ROS production and DNA damage [13,14]. At the cellular level, exposure to cold stress can reduce cell membrane fluidity and alter membrane phospholipid composition [15,16]. In addition, intracellular enzyme activity, respiration, and metabolic rates are reduced upon exposure to cold stress [17,18]. The global adverse effects of cold stress on biomolecules and cells ultimately result in the death of the organism.

Under selective pressures during evolution, fishes have developed biological mechanisms to cope with the challenges caused by confronting cold stress. The mechanisms determining the cold resistance of fish have attracted intensive research interests due to the great theoretical and application significances. Lots of transcriptomics [19,20,21,22,23,24,25,26,27], proteomics [28,29,30,31,32], metabolomics [33,34], and integrated-omics [35] investigations have been conducted to explore the mechanisms underlying cold acclimation and the limit of cold endurance in multiple fish species. Numerous genes, proteins, and pre-mRNA splicing and protein post-translation modification events were revealed to associate with the cold resistance of fish. Furthermore, functions of several key biological pathways enriched among the identified genes, proteins, and metabolites in regulating the cold resistance of fish were confirmed using specific inhibitors [23,36], and mutant fish lines generated through gene knockout [37] or insertional mutagenesis [38].

Autophagy and ER stress response are among the pathways enriched for cold-induced genes, which regulate the cold resistance of fish through maintaining cell homeostasis. Autophagy is a self-degradation process whereby long-lived proteins, damaged organelles, and even invasive pathogens are engulfed, degraded, and reused to protect the cells against disturbances caused by various stresses [39]. The exposure of zebrafish larvae to lethal cold stress and rewarming induced multiple autophagy genes, such as *atg2a*, *atg2b*, *smcr8a*, and *smcr8b*, and the inhibition of autophagy using Bafilomycin A1, significantly sensitized the fish to cold stress [23]. It was reported that fasting can enhance the cold resistance of zebrafish through stimulating lipid catabolism and autophagy, and the knockout of the autophagy gene *atg12* and treatment with the autophagy inhibitor chloroquine both significantly decreased the survival rate of fish under cold stress [37]. These studies substantiate the roles of autophagy in enhancing the resilience of fish to cold stress. ER stress response, also known as unfolded protein response (UPR), is a defense mechanism aimed at restoring ER homeostasis for cell survival [40]. The protective effect of hypothermia preconditioning for human cortical neurons against lethal cold injury was ascribed to the activation of UPR [41]. However, a prolonged ER stress response can activate apoptotic signals [42] and is implicated in cold-induced cell apoptosis and tissue dysfunction [38,43].

Due to the critical roles of autophagy and UPR in protecting cells against cold stress, delineating the new factors regulating their activities is helpful to understand the genetic basis determining the resistance to cold of fish. An ER-localized transmembrane protein, TMEM39A, was previously reported to regulate both ER stress response and autophagy [44,45]. TMEM39A also plays functions independent of ER stress response and autophagy, such as mediating collagen secretion [44], regulating lysosome distribution, and lysosome-associated signaling [46]. Genetic variants in the human *TMEM39A* gene are related to multiple autoimmune diseases [47]. *TMEM39B* is the paralog of *TMEM39A*, which only exists in the genome of vertebrates. Despite the functions of *TMEM39A* being well characterized, those of *TMEM39B* remain completely unknown.

To characterize the molecular functions of Tmem39b in zebrafish, we generate *tmem39b*-knockout lines and test whether the elimination of this gene would result in decreased cold resistance through impairing ER stress response and autophagy. We observe that the dysfunction of Tmem39b significantly reduces survival rates and exacerbates the tissue damage of the fish exposed to lethal cold stress. Transcriptomic analyses reveal that Tmem39b plays multifaceted functions in regulating the cold resilience of fish. The deficiency of Tmem39b attenuates both cold- and rewarming-activated transcriptional programs. The consequences of these effects on gene transcription are reflected by exaggerated DNA damage during cold exposure and impaired damage repair upon recovery at normal temperature. The gene modules affected by the *tmem39b* mutation are highly enriched in immune-related biological processes and pathways. However, a relatively minor effect is identified for *tmem39b* in cold- or rewarming-induced ER stress responses and autophagy based on the transcriptional expression of the genes involved in these pathways. Our results provide novel insights into the molecular functions of Tmem39b and link the cold resilience of fish with tissue damage repair and activation of immune responses.

## 2. Results

### 2.1. Molecular Characteristics of Zebrafish Tmem39b

The structure of the zebrafish Tmem39b protein was predicted for a better understanding its functions. Zebrafish Tmem39b consists of eight transmembrane domains, and most parts of the protein, including both the N and C terminals, are located inside the membrane (Figure 1A). A phylogenetic tree was generated for the TMEM39 proteins of species, including human (*Homo sapiens*), chicken (*Gallus gallus*), zebrafish, fruit fly (*Drosophila melanogaster*), and *C. elegans*. Consistent with the previous report [46], the clade of vertebrate TMEM39B was clearly separated from that composed of vertebrate TMEM39A and invertebrate TMEM39 (Figure 1B). Furthermore, a sequence alignment analysis revealed that the sequence identity among TMEM39A or TMEM39B from different species was significantly higher than that between TMEM39A and TMEM39B from the same species (Figure 1C), suggesting a considerable functional differentiation between these two paralogs.

Since TMEM39A was reported to be an ER-localized transmembrane protein, we tested whether zebrafish Tmem39b was localized to ER by co-transfecting HeLa cells with pAc-Tmem39b-GFP and pDsRed-ER (an ER marker expressing plasmid). The GFP-tagged Tmem39b was observed to be co-localized with the ER-DsRed (Figure 1D), indicating that zebrafish Tmem39b was also an ER-localized transmembrane protein. Moreover, the tissue distribution pattern of *tmem39b* transcripts was explored by quantitative real-time PCR (qPCR). The results demonstrate that zebrafish *tmem39b* is extensively expressed in the tissues of adults. The highest transcription level was detected in the ovary, followed by the intestine and brain (Figure 1E).

### 2.2. Deficiency of Tmem39b Sensitizes Zebrafish Larvae to Lethal Cold Stress

To investigate the functions of zebrafish Tmem39b in regulating cold resistance, the gene knock-out fish lines were first generated using the CRISPR/Cas9 system. Two knockout zebrafish lines were obtained and submitted to the China Zebrafish Resource Center (CZRC). The accession numbers of the mutant lines were zko3151a (3151a, −28 bp) and zko3151b (3151b, −11 bp), respectively. The genotypes of the mutant lines were confirmed using DNA sequencing. The mutations result in the frame shift and truncation of the encoded protein (Figure 2A). The mutants were viable and fertile, and demonstrated no defects in embryogenesis and early development (Appendix A). Interestingly, adult males of zko3151b had a significantly lower standard length (*p* < 0.01) and body weight (*p* < 0.01) in comparison to the wild-type (WT) males (Appendix A).

The zko3151b larvae were exposed to lethal cold stress (10 °C) to verify the function of Tmem39b in regulating the cold resistance of zebrafish. The cold sensitivity assay was repeated using the larvae of multiple generations. The mutant larvae consistently demonstrated significantly lower survival rates under lethal cold stress in comparison to the matched WT (Figure 2B,C). These results indicate that Tmem39b plays an essential role in the resilience of zebrafish larvae to lethal cold stress.

### 2.3. Mutation of Tmem39b Exacerbates Cold-Induced Tissue Damage in Adult Zebrafish

Cold resistance was also compared between adults of the mutants and the WT. The survival rate of the zko3151b adults under lethal cold stress was significantly lower than that of the WT (*p* < 0.001, Figure 3A). Furthermore, the adults generated by crossing heterozygous females and males of zko3151a were also characterized. This population contains three genotypes, including homozygote (aa), heterozygote (Aa), and wild type (wt). In agreement with the results of the zko3151b mutants, the survival probability of the homozygous zko3151a mutants is significantly less than that of the WT (*p* < 0.001), while no significant difference is observed between the heterozygotes and WT (Appendix A).

Histological analyses for tissues, including the brain, gill, liver, and intestine, were performed to assess the functions of Tmem39b in maintaining tissue homeostasis upon exposure to lethal cold stress. When compared to the untreated controls (WT-28 °C), exposure to lethal cold stress elicited obvious structural changes in the tissues of both the WT (WT-10 °C) and mutant fish (zko3151b-10 °C, Figure 3B). Cold exposure resulted in vacuoles in the brain and liver, and destroyed the tissue integrity of the gill and intestine (Figure 3B). Tissues of the mutants demonstrated more severe damage in comparison to those of the WT, as the high number and large size of vacuoles in the brain and liver and more severe cell exfoliation for the gill lamella and intestinal microvilli were observed in the mutants (Figure 3B).

In accordance with the results of the histological analyses, following cold exposure, the intensities of the TUNEL signal in the tissues of the mutants were significantly greater than those of the WT (Figure 3C). Consistently, more apoptotic cells were detected in tissues obtained from the mutants (Figure 3D). While the apoptotic cells were scattered in the brain and liver sections, those of the intestine were concentrated in the microvilli (Figure 3D). These results indicate that zebrafish Tmem39b plays an important role in protecting the organism from cold-induced tissue damage.

### 2.4. Effects of Tmem39b Mutation on Gene Transcription upon Cold Stress and Rewarming

To explore the molecular mechanisms underlying the functions of zebrafish Tmem39b in regulating cold resistance, the effects of *tmem39b* mutation on gene transcription were characterized through RNA-seq. Both the zko3151b and WT larvae were exposed to cold stress (cs, treated at 10 °C for 12 h) or rewarming (re, treated at 10 °C for 12 h followed by recovery at 28 °C for 6h). The untreated controls (ctrl, developed to 96 hpf under 28 °C) were also included in the analysis. Four biological replicates were included for each treatment and a total of 24 libraries were generated and sequenced.

The results of principle component analysis (PCA) reveal an obvious separation between the zko3151b-re and WT-re samples, while the ctrl and cs samples of the mutants are clustered together with those of WT (Figure 4A), suggesting severe effects of the *tmem39b* mutation on gene transcription during the rewarming phase. Numbers of the differentially expressed genes (DEGs) between different experimental groups are displayed in Figure 4B. The changes between treatment conditions elicited a high number of DEGs in both the mutant and WT larvae. For example, the exposure of the WT larvae to cold stress resulted in 2333 up-regulated (U1) and 2129 down-regulated (D1) genes; while recovery at 28 °C caused the up-regulation of 3350 (U2) and down-regulation of 2917 genes (D2), respectively (Figure 4C). DEG numbers for the zko3151b mutants were always slightly higher than those of the WT under the same condition. In agreement with the PCA results, the highest numbers of both up- and down-regulated genes between the mutant and WT were observed during the rewarming phase.

The Venn analyses identified genes regulated by both the cold-warm stress and *tmem39b* mutation. While the mutant shared most of the up- (1854) and down-regulated (1521) genes with WT during cold-stress exposure, considerable numbers of cold- or rewarming-responsive genes were observed to be influenced by the *tmem39b* mutation (Figure 4C). A total of 126 cold-induced (GS1) and 94 cold-inhibited genes (GS2) were observed to be differentially expressed between the zko3151b-cs and WT-cs groups (Figure 4C). In other words, the up- or down-regulation of these genes in cold-stress conditions was attenuated or reversed by the *tmem39b* mutation. The numbers for the genes regulated by both rewarming and *tmem39b* mutation were 302 (GS3) and 182 (GS4), which are double of the cold-regulated ones (Figure 4C), suggesting greater effects of Tmem39b in shaping the transcriptome in the rewarming phase. The genes of GS3 and GS4 were overlaid on those differentially expressed between the 3151b-re and WT-re groups; one of the GS3 genes, *crp2*, was highlighted (Figure 4D). The genes of GS1-GS4 are listed in Appendix A.

### 2.5. Dysfunction of Tmem39b Affects Both Cold- and Rewarming-Responsive Genes

Functional enrichments for the cold- and rewarming-responsive genes that were also regulated by Tmem39b dysfunction are displayed in Appendix A. Although no enrichments were observed for the genes of GS2 and GS4, the genes of GS1 were enriched in functional terms, including nucleosome and chromatin (cellular component), chromatin/chromatin-binding, or -regulatory protein (protein class). The genes of GS3 are highly enriched in biological processes, such as immune response, defense response, and response to stimulus.

The genes of GS1 and GS3 were ranked according to the significance (adjust *p*-value) of differential expression, and the top ones are displayed in Figure 5A,B, respectively. Genes of GS1, such as *fosab* and *egr1*, are commonly identified cold-induced transcription factors, which are implicated in regulating the establishment of cold resistance [19,20,23]. Their transcriptional expressions were highly induced by cold stress, while the up-regulation was significantly attenuated in the zko3151b mutants (Figure 5A). Genes, such as *c3a.3*, *il11a*, *mmp9*, and *sting1*, are the representatives of GS3. These genes are associated with immune functions. Their up-regulation was exclusively detected in the rewarming phase and again dependent on the function of Tmem39b (Figure 5B). Tmem39b-dependent up-regulation of the representative genes belonging to GS3 (*bif1.2*, *fosab*, *rbp7a*, and *socs3a*) and GS4 (*c3a.3*, *il11a*, *mmp9*, and *sting1*) was also confirmed by qPCR assays (Figure 5C). Simultaneously, the loss of *tmem39b* leads to considerable effects on the expression of both cold- and rewarming-responsive genes.

### 2.6. Tmem39b Protects Zebrafish from Cold-Warm Stress-Induced DNA Damage

Based on the fact that zebrafish Tmem39b affects the transcription of cold-induced transcription factors and rewarming-activated immune genes, we postulated that Tmem39b has a role in both the establishment of cold resistance and the repair of cold-caused tissue damage. To test this hypothesis, we measured the level of phosphorylated H2AX at Ser139 (γH2AX), a sensitive marker for DNA damage [48], as a proxy for cold-warm stress-caused cellular damage. For the WT larvae, exposure to cold stress slowly increased the level of γH2AX; however, rewarming at 28 °C rapidly induced the content of γH2AX and the highest level was observed at 2 h. Subsequently, the γH2AX content gradually decreased to basal level (Figure 6). These results indicate that rewarming following cold exposure exacerbated the degree of DNA damage and the damage could be gradually repaired during recovery at a normal temperature.

Upon cold exposure, no significant difference in the γH2AX content was identified between the zko3151b mutants and WT until 12 h; however, the γH2AX content of the mutants abruptly increased at 24 h and was obviously greater than that of the WT. Despite showing the same changing pattern, tissue γH2AX contents of the mutants were significantly greater than those of the WT during the recovery process (Figure 6). Together, these results indicate that the mutation of *tmem39b* sensitized the fish to cold-warm stress-induced DNA damage and delayed the repair of the damage during recovery under a normal temperature. Therefore, zebrafish Tmem39b plays a key role in enhancing cold resilience and facilitating tissue-damage repair.

Since the reactive oxygen species (ROS) is the main cause of damage to large molecules and fish tissues upon cold exposure [21,49], we measured the tissue ROS content of the mutant and WT larvae exposed to cold-warm stress. Consistent with the results of γH2AX detection, the tissue ROS level only slightly increased upon cold stress, but sharply rose during the recovery phase. Although γH2AX was reduced to the basal level during recovery, the level of ROS did not decrease (Appendix A). Furthermore, no difference in the ROS content was identified between the mutants and WT under cold-warm stress (Appendix A), indicating that the functions of Tmem39b are independent in the generation and clearance of ROS.

### 2.7. Effects of Tmem39b Dysfunction on ER Stress Response and Autophagy

Both ER stress response and autophagy are important for enhancing the resistance of fish to the cold [23,37,38,50]. TMEM39A, the paralog of TMEM39B, is involved in the regulation of ER stress response and autophagy [44,45]. We analyzed the transcriptomic data to verify whether the mutation of zebrafish *tmem39b* affects the transcription of genes involved in ER stress response and autophagy. Our previous dynamic transcriptomic data of zebrafish larvae exposed to cold-warm stress revealed the genes involved in ER stress response and autophagy [23]. The ER stress response-associated genes (Figure 7A, presented in red) were classified into two clusters: genes including *hspa5*, *atf4a*, and *xbp1* were induced following exposure to cold stress for more than 6h, while those, such as *derl1*, *sec63*, and *atf6* were exclusively up-regulated upon rewarming. The up-regulation of genes involved in autophagy (Figure 7A, presented in blue) usually initiated after 24 h of cold exposure and lasted for 12 h upon recovery at 28 °C.

The loss of *tmem39b* demonstrated no obvious effects on the transcriptional expression of the genes involved in ER stress response and autophagy (Figure 7A). The expression of representative genes in the mutant and WT larvae upon cold-warm stress was also analyzed with qPCR. The expressions of *atf4a* and *xbp1* in the mutants upon rewarming were significantly lower than in the WT (Figure 7B) and no significant difference in the expression of autophagy-associated genes was observed between the mutant and WT (Figure 7C). These results indicate that the mutation of *tmem39b* has relatively minor effects on ER stress response and autophagy.

### 2.8. Gene Modules Associated with Functions of Tmem39b in Regulating Cold Resistance

The DEGs were classified into 15 clusters through K-means clustering to identify the gene modules associated with functions of Tmem39b in regulating stress resilience. Among the gene clusters, C3, C4, C8, C10, C11, C13, and C14 were responsive to cold-warm stress and affected by *tmem39b* mutation; C7 and C15 were exclusively influenced by *tmem39b* mutation; and the others were only responsive to cold-warm stress (Figure 8A). The eigengenes for modules C3, C10, C13, and C14 were calculated to further illustrate their differential expressions between the mutants and WT under different conditions. Genes of C3 and C13 were mainly differentially expressed between 3151b-cs and WT-cs, while those of C10 and C14 were differentially expressed between 3151-re and WT-re (Figure 8B).

GO and KEGG pathway-enrichment analyses were performed for these gene modules to uncover the biological functions regulated by Tmem39b. GO enrichments (biological process) are displayed in Appendix A and the top ones are presented in Figure 8C. The most prominent result was observed for C14, which demonstrates multiple highly significant GO enrichments associated with immune responses, such as humoral immune response and the regulation of immune system response. The genes of this module were highly up-regulated during the recovery stage and the mutation of *tmem39b* attenuated their up-regulation (Figure 8A,B). Genes of C7, C8, and C10 were also enriched in immune processes, such as the positive regulation of T-cell migration, lymphocyte migration, and neutrophil activation. Basal or stress-induced expressions of genes in C7, C8, and C10 were also negatively affected by *tmem39b* mutation. These results indicate that zebrafish Tmem39b plays an essential role in potentiating the basal and stress-induced immune responses.

The results of the KEGG pathway-enrichment analyses are presented in Appendix A and the top terms are displayed in Figure 8D. Based on the gene expression data, pathways enriched for the genes of C3 (p53 signaling, necroptosis, and apoptosis) and C14 (mitophagy, NOD-like receptor signaling, and cytokine-cytokine receptor interaction) were the most likely candidates associated with the functions of Tmem39b in regulating cold resistance. The up-regulation of the genes involved in these pathways upon cold exposure (C3) and recovery (C14) were dependent on the functional Tmem39b.

### 2.9. Dysfunction of Tmem39b Attenuates the Induction of CRP by Rewarming

To provide further evidence for the roles of Tmem39b in regulating stress-induced immune functions, we measured the concentration of CRP of both the mutant and WT larvae exposed to cold-warm stress. CRPs are secreted immune factors, which function as pattern-recognition receptors to bind the pathogens and apoptotic cells to facilitate their clearance by phagocytes [51]. The up-regulation of the *crp2* gene by rewarming following cold exposure was highly Tmem39b-dependent (Figure 4D). Consistent with the transcriptomic data, only the WT larvae demonstrated a significant induction of tissue CRP concentration upon rewarming at 28 °C (Figure 9A). These results suggest that Tmem39b plays an important role in mediating production and secretion and the CRPs.

## 3. Discussion

ER stress response and autophagy are the key pathways enhancing the cold resistance of fish through eliminating damaged cellular components and restoring cell homeostasis upon exposure to lethal cold stress. Based on functions of TMEM39A in regulating ER stress response and autophagy, we postulated that Tmem39b, the vertebrate-specific paralog of TMEM39A, may play an important role in regulating the cold resilience of fish through similar mechanisms. To test this hypothesis, we generated *tmem39b*-knockout zebrafish lines and compared their survival ability under lethal cold stress with that of the WT fish. We observed that a loss of *tmem39b* decreased the survival rates of zebrafish upon cold-warm stress and aggravated tissue damage and cell apoptosis under continuous cold exposure. These results indicate that Tmem39b plays a critical role in regulating the cold resistance of zebrafish.

We performed RNA-seq to identify the genes and biological pathways affected by *tmem39b* mutation to provide insights into the molecular mechanisms underlying functions of Tmem39b in regulating the cold resistance of zebrafish. We used the cold-rewarming regime to test the cold sensitivity of zebrafish larvae, and thousands of genes were previously observed to be differentially expressed during the rewarming phase following exposure to the cold [23]. Therefore, the genetic programs activated upon rewarming also contributed to the cold resilience of the fish. The results of RNA-seq reveal that a loss of *tmem39b* affects the expression of both cold- and rewarming-regulated genes. We hypothesized that zebrafish Tmem39b plays a pivotal role in regulating both the establishment of cold resistance upon exposure to hypothermia stress and the restoration of cellular homeostasis during the recovery phase following exposure to the cold.

We measured the level of γH2AX, an established DNA damage marker [48], as an index for tissue damage caused by cold-warm stress. The results indicate that rewarming following exposure to the cold further exaggerated the DNA damage of the cells, and the damage caused by cold-warm stress could be gradually repaired upon recovery at a normal temperature. ROS production was suggested to be the main cause of damage to macro-biomolecules, such as DNA upon exposure to cold stress [52]. In accordance with the changing patterns in γH2AX levels, exposure to cold stress also induced ROS production, and rewarming further enhanced ROS concentration. The induction of ROS content and DNA damage upon cold-warm stress resembled cold ischemia-reperfusion injury that occurs during a human solid-organ transplantation [53]. These data suggest that the genetic programs activated by both cold stress and rewarming are involved in determining the cold resilience of fish. The former is mainly associated with enhancing cold resistance, while the latter is responsible for the reparation of cellular damage caused by stress. The loss of *tmem39b* aggravated DNA damage during cold-warm stress and delayed damage reparation during the recovery phase, indicating the multifaceted function of Tmem39b in regulating the cold resilience of fish.

The functions of zebrafish Tmem39b in regulating cold resistance were independent of the generation and clearance of ROS, since the mutation of *tmem39* showed no significant effects on ROS levels upon cold exposure and rewarming. Because zebrafish Tmem39b is also localized to the ER membrane and TMEM39A regulates both ER stress response and autophagy [44,45], we checked the expression of ER stress- and autophagy-associated genes that were up-regulated by cold-warm stress in the zko3151b mutants. Only a minor effect was identified for *tmem39b* mutants on the expression of the representative genes. These results suggest that functions of Tmem39b are largely independent of ER stress response and autophagy.

The mutation of *tmem39b* demonstrated greater effects on the rewarming-responsive genes than the cold-regulated ones based on PCA results and DEG numbers. The genes induced by rewarming and simultaneously regulated by Tmem39b deficiency were highly enriched in the biological processes associated with immune responses. The up-regulation of immune-related genes, such as *c3a.3*, *il11a*, *mmp9*, and *sting1*, by rewarming was highly dependent on the functions of Tmem39b. These results support the function of zebrafish Tmem39b in regulating immune responses elicited by cold-warm stress. Despite the activation of immune responses that have not been directly linked with the enhancement of the cold resistance of fish, it is well known that immune responses can be activated by damage-associated molecular pattern (DAMP) molecules released from damaged and dying cells and play an important role in damage repair and tissue regeneration [54,55,56]. The deregulation of the dedicated immune responses to tissue damage can ultimately lead to organ failure and organism death [55].

We observed that many genes inhibited by *tmem39b* mutation encoded secreted immune proteins, such as *crp2*, *mpeg1.2*, *itln2*, and *zgc:171497* (Appendix A). Therefore, we hypothesized that zebrafish Tmem39b exerted its function in immune regulation through mediating the secretion of these immune factors, similar to its paralog TMEM39A, which regulates autophagy through controlling the trafficking of the PtdIns(4)P phosphatase SAC1 [45]. It can be reasoned that the deficiency of Tmem39b may result in the accumulation of target proteins in the ER lumen, thus activating unknown mechanisms to inhibit the transcription of the corresponding genes. To corroborate the activity of Tmem39b in regulating immune functions, we measured the CRP concentration in both the mutant and WT zebrafish larvae exposed to cold/warm stress, since *crp2*, one CRP coding gene, was inhibited by *tmem39b* mutation. The results indicate that the dysfunction of Tmem39b dampens the up-regulation of tissue CRP content upon rewarming. CRPs are critical immune mediators that can bind to the apoptotic and necrotic cells and promote their clearance by macrophages [51].

Thus, Tmem39b appears to enhance the cold resilience of zebrafish through orchestrating cold-induced transcriptional programs and the immune responses to facilitate the establishment of cold resistance upon cold exposure and repair of stress-induced tissue damage during rewarming (Figure 9B). The molecular mechanisms underlying zebrafish Tmem39b’s activities in regulating these biological processes remain to be further explored.

## 4. Materials and Methods

### 4.1. Experimental Fish

Adult zebrafish of the AB line were maintained in a circulating water system, as previously described [23]. The fish room was illuminated with 12 h/12 h light/dark cycles (8:00 a.m. to 8:00 p.m.). The water temperature was controlled at 28 ± 1 °C. Water quality parameters, including pH (7.0–7.5), dissolved oxygen (>5 mg/L), and ammonia (<0.01 mg/L), were frequently measured. Artificial fertilization and egg cultivation were performed following our previous protocol [23]. The embryos and larvae of zebrafish were maintained at 28 °C using a biochemical incubator (HWS-150, Shanghai Jinghong laboratory instrument Co., Ltd., Shanghai, China).

### 4.2. Molecular Cloning, Cell Culture, and Transfection

Total RNA was isolated from 96 hpf zebrafish larvae using TRIzol™ (Thermo Fisher, Waltham, MA, USA) and first-stranded cDNA was synthetized using the RevertAid First Strand cDNA Synthesis Kit (Thermo Fisher, Waltham, MA, USA). The open reading frame (ORF) of zebrafish *tmem39b* (GenBank accession number: NM_199860.1) was amplified using primers *tmem39b-cds-F*/*tmem39b-cds-R* (Appendix A) and cloned into the pAcGFP-N1 vector (Clontech, CA, USA) between the XhoI and BamHI restriction enzyme sites to generate the pAc-Tmem39b-GFP construct. The sequence was confirmed by DNA sequencing. The protein structure of zebrafish Tmem39b was predicted by Protter [57]. Protein sequence alignment was conducted using clustalx (v-2.1) [58] and a phylogenetic tree for the TMEM39 proteins of multiple species was generated by MEGA (v-11) [59] using the neighbor-joining method.

To characterize the subcellular localization of zebrafish Tmem39b, the pAc-Tmem39b-GFP construct was used to transfect HeLa cells (ATCC number CCL-2). The cells were cultured in DMEM/High glucose (Hyclone, Utah, USA) supplemented with 10% fetal bovine serum from PAN-Biotech (Aidenbach, Germany) and were incubated at 37 °C using an incubator (Thermo Fisher) supplied with 5% CO_2_. One day before transfection, the cells were seeded onto 35 mm Petri dishes at a density of 2 × 10^5^ cells per dish. The culture medium was replaced with fresh culture medium 1h before transfection. The cells were co-transfected with 1 μg pAc-Tmem39b-GFP (or pAcGFP-N1) and 1 μg pDsRed-ER (obtained from Clontech, CA, USA) using 1 μL VigoFect (Vigorous Biotech, Beijing, China). The culture medium was changed at 6h following transfection. At 24h following transfection, the cells were washed twice with PBS and fixed with 4% PFA (obtained from Beyotime, Shanghai, China) for 1 h. After being washed three times with PBS, the cells were stained with DAPI-staining solution (BOSTER, Wuhan, China). Finally, the cells were washed and observed with an SP8 confocal microscope from Leica (Wetzlar, Germany). Photos were taken under a 63 × oil immersion objective.

### 4.3. Real-Time Quantitative PCR

Real-time quantitative PCR (qPCR) was conducted to detect the relative gene expression. To characterize the transcriptional expression pattern of *tmem39b* in adult zebrafish, the fish were dissected after being anesthetized with 160 mg/L MS-222 (Sigma, St. Louis, MO, USA). Tissues, including brain, eye, gill, heart, intestine, kidney, liver, muscle, ovary, spleen, and testis, were collected and subjected to total RNA extraction as previously described. The *18s rRNA* was used as an internal control to normalize the expression levels of *tmem39b* among tissues. The expression of representative cold/rewarm stress-regulated genes was also analyzed using qPCR, and two internal controls (*mdh2* and *slc25a5* genes) were used to normalize the expression of these genes. A Bio-Rad CFX Duet Real-Time PCR System (Hercules, CA, USA) was used for the qPCR assays. The sequences, amplification size, and efficiency of the primers used for qPCR are listed in Appendix A. The reagents, program, and method for the data analysis of qPCR were the same as our previous study [23].

### 4.4. Generation of Gene Knockout Zebrafish Lines

The CRISPR/Cas9 system was utilized to generate gene knockout zebrafish lines. The template of single-guide RNA (sgRNA) was generated using a cloning-free protocol as previously reported [60]. Briefly, the online software CHOPCHOP [61] was used to obtain the highly specific and efficient target sites. The adopted target sequence for the *tmem39b* gene was 5′-GGGCATGTCCAGTCCACCGT-3′, located in the third exon of the gene. The targeting oligo consisting of the T7 promoter, target sequence, and 20-nt sequence matching a generic sgRNA template was synthesized by Sangon Biotech (Shanghai, China). The targeting oligo was annealed with the 80-nt chimeric sgRNA core sequence [60]. The annealed oligos were filled in using the I-5™ 2X Hi-Fi PCR Master Mix from MCLAB (South San Francisco, CA, USA) under the following program: 98 °C for 2 min; 50 °C for 10 min; and 72 °C for 10 min. The product was purified and used as a template for the subsequent transcription reaction. The sgRNA was synthesized using the TranscriptAid T7 High-Yield Transcription Kit (Thermo Fisher, Waltham, MA, USA) following the manufacturer’s instructions.

The sgRNA was purified, diluted, and mixed with Cas9 protein (EnGen^®^ Spy Cas9 NLS) obtained from New England Biolabs (NEB). The final concentration of sgRNA and Cas9 protein were 100 ng/μL and 5 μM, respectively. The mix was incubated at room temperature for 20 min to generate Cas9 RNP complexes. Zebrafish embryos were injected with the Cas9 RNP solution (1–2 nL/embryo) at the single-cell stage using a PICO-LITER injector from WARNER (Holliston, MA, USA), as previously described [62]. At 48 h post-fertilization (hpf), 10 injected embryos were randomly selected for the cutting efficiency test. The embryos were lysed in 100 μL 50 mM NaOH at 95 °C for 10 min, then 10 μL 1 M Tris-Cl (pH 8.0) was added to neutralize the solution. The genomic region surrounding the target site was amplified through PCR using the primers gtmem39b-F/gtmem39b-R (Appendix A) and the embryo lysate as a template. After the completion of the amplification, the 537-bp PCR products were subjected to a re-annealing process to enable hetero duplex formation [63]. The final product was digested with T7 Endonuclease I (T7E1, NEB) and positive results were judged by the cleavage of the hetero duplex at the mismatch site. Successful editing of the target gene was further confirmed by DNA sequencing and overlapping peaks could be observed around the target site.

The remaining F0 embryos were cultivated to sexual maturation and individually crossed with WT fish. The F1 embryos obtained for each F0 fish were subjected to the T7E1 assay as described above. The litters past the test were retained and raised to adults. The F1 fish were individually genotyped through the T7E1 assay and DNA sequencing. The positive ones were crossed with WT fish to generate the F2 offspring. The F2 fish were genotyped and the positive male and female fish were intercrossed to generate the F3 offspring. Homozygous fish lines with different genotypes were established through selecting homozygous males and females from the corresponding F3 offspring.

### 4.5. Cold-Sensitivity Assays

To compare the cold resistance of the *tmem39b* mutant and WT larvae, larvae cultured to 96 hpf under 28 °C were exposed to acute cold stress as previously reported [23]. Briefly, the larvae were randomly assigned into 60 mm Petri dishes (40 individuals per dish) and immediately exposed to 10 °C preconditioned E3 medium (5 mM NaCl, 0.17 mM KCl, 0.33 mM CaCl_2_, and 0.33 mM MgSO_4_, pH 7.2). After being maintained at 10 °C for 12, 18, 24, and 30 h, the fish were returned to 28 °C and incubated for another 24 h. The fish were frequently checked during the recovery phase and the dead ones were removed and recorded. At the end of the experiment, the fish were anesthetized with 160 mg/L MS-222 and all counted. Four biological replicates were included for each treatment. To confirm the cold sensitivity of the *tmem39b* mutants, larvae of different generations were tested. Photographs of the cold-treated larvae were taken using a Zeiss (Oberkochen, Germany) stereomicroscope equipped with a color CCD camera.

To compare the cold resistance of the adults, embryos of the *teme39b* mutant and WT were collected in the same day and raised to 4 months of age under the same condition. After being fasted for 1 d, the fish were taken out of the raising system and randomly assigned into exposure baskets (16 cm × 16 cm × 25 cm) made of porous stainless-steel sheets (thickness: 1 mm; pore diameter: 3 mm). The baskets were divided into two parts at the middle, so the mutant and WT fish could be treated side by side. A plastic box (chamber size: 60 cm × 50 cm × 35 cm) was used to hold the steel baskets (up to 6). A PC200 A40 ARCTIC Refrigerated Circulator (Thermo Scientific) was used for temperature control. Two mini pumps (40 W, 2000 L/h) were used to exchange the water between the plastic box and chamber of the refrigerated circulator. The fish were acclimated to the experimental condition for 1 d at 28 °C. In the morning of the second day, the water temperature was decreased from 28 to 18 °C with a constant rate (2 °C/h) and then maintained. The water temperature was decreased further from 18 °C to 10 °C with the same rate in the morning of the third day. Then, the water temperature was maintained at 10 °C. The fish were checked every 12 h and the dead ones were taken out and their survival time under 10 °C was recorded. Thirty fish for each line were included in the experiment.

### 4.6. Histological and TUNEL Assays

To explore the effects of cold exposure on tissue structure, adults of both the WT and 3151b mutant were exposed to 10 °C for 2 d and the survivors were dissected. Tissues, including the brain, gill, liver, and intestine, were collected. Sample fixation, paraffin section, and hematoxylin and eosin (H&E) staining were performed as previously described [64]. The tissues obtained from an untreated WT fish were also included. Images for the sections were captured using an Aperio VERSA Brightfield, Fluorescence & FISH Digital Pathology Scanner from Leica (Wetzlar, Germany).

TUNEL (terminal deoxynucleotidyl transferase-mediated dUTP nick-end labeling) was performed to detect apoptotic cells in the tissues. The same batch of tissue sections used for H&E staining was subjected to the TUNEL assay. Briefly, the sections were dewaxed in xylene, rehydrated with graded ethanols, and treated with proteinase K solution (20 μg/mL, dissolved in 10 mM Tris-Cl, pH 7.4) from Promega (Madison, WI, USA). After being thoroughly washed with PBS, apoptotic cells were detected using the One-Step TUNEL Apoptosis Assay Kit from Beyotime (Shanghai, China), according to the manufacturer’s instructions. Photos for the sections were taken using a Leica SP8 confocal microscope (Wetzlar, Germany). The pictures were analyzed with ImageJ2 (National Institute of Health, Bethesda, MD, USA) [65] for the quantification of the fluorescence intensity (TUNEL signal).

### 4.7. RNA-Sequencing and Data Analysis

To prepare samples for RNA-sequencing, both the WT and 3151b mutant zebrafish larvae developed to 96 hpf under 28 °C were exposed to cold shock at 10 °C for 12 h (designated as cs), rewarmed at 28 °C for 6h after 10 °C exposure for 12 h (re), and the untreated larvae at 96 hpf were used as the control (ctrl). The procedures for cold exposure and rewarming were the same as cold-sensitivity assays for the larvae. Four replicates were included for each treatment. At the end of exposure, the dead fish were removed (if any) and pieces of ice were added into the dishes for immediate anesthetization of the fish. Total RNA extraction and concentration measurements were conducted as previously described [23]. The samples were subjected to quality analysis, library construction, and high-throughput sequencing by the Biomarker Technologies (Beijing, China). The RNA-seq data of this study were deposited in the NCBI Sequence Read Archive (SRA) with the BioProject accession number PRJNA857857.

The raw reads were preprocessed by fastp-v0.23.2 [66] using default parameters to filter low-quality reads. The clean reads were mapped to the cDNA sequences of zebrafish (downloaded from Ensembl http://asia.ensembl.org/, release-105 (accessed on 15 October 2021)) using Salmon-v1.6.0 [67] with parameters “--gcBias --seqBias --posBias --validateMappings”. The results of Salmon were processed by tximport [68] to summarize gene transcriptional abundance (expressed as transcript per million, TPM) and to generate the raw number of mapped reads for each gene. Genes with a TPM value ≥ 1 in all the biological replicates for at least one treatment group were regarded as expressed, and only the genes pasted this criterion were subjected to subsequent analyses.

The differentially expressed genes (DEGs) between various experimental groups were identified using DESeq2-v1.30.0 [69] with the following thresholds: Foldchange ≥ 1.5 and adjusted *p*-value ≤ 0.05. Principle component analysis (PCA) for the gene expression data was performed using ArrayTrack [70]. Heat maps for the gene expression data and Venn charts for different gene sets were generated using GENE-E (https://software.broadinstitute.org/GENE-E (accessed on 21 September 2022)) and Venny-v2.1.0 (https://bioinfogp.cnb.csic.es/tools/venny (accessed on 21 September 2022)), respectively. K-means clustering of all the DEGs were conducted using cluster 3.0 [71]. To visualize the overall expression pattern of genes among different experimental groups, the abundance of genes belonging to the interested clusters was analyzed using the WGCNA package [72] to generate the eigengene values. GO (gene ontology) and KEGG (Kyoto Encyclopedia of Genes and Genomes) pathway enrichment analyses were performed using PANTHER-v17.0 (http://www.patherdb.org/) [73] and ClueGO-v2.5.8 (National Institute of General Medical Sciences, Bethesda, MD, USA) [74], a plugin of Cytoscape-v3.9.1 (Cytoscape Consortium, San Diego, CA, USA) [75].

### 4.8. Western Blotting

Western blotting was performed, as previously described [23], to detect the level of phosphorylated gamma H2A.X (S139). Both the WT and 3151b larvae (at 96 hpf) were exposed to 10 °C for 1, 2, 6, 12, and 24 h, or treated at 10 °C for 12 h followed by recovery at 28 °C for 1, 2, 6, 12, and 24 h as described above. At the indicated time points, a trichloroacetic acid solution (final concentration 4%) was added into the plates to immediately terminate life activity of the fish. The samples were collected in 1.5 mL centrifuge tubes, washed twice with pre-chilled acetone, and once with PBS. Subsequently, the samples were homogenized in SDS Lysis Buffer from Beyotime (Shanghai, China) supplemented with protease inhibitor cocktail (1:100) and PMSF (1:100); both were obtained from Beyotime (Shanghai, China). The lysate was centrifuged at 4 °C, 12,000× *g* for 10 min. The supernatant was collected and subjected to a protein-concentration measurement using a BCA Protein Assay Kit (Beyotime, Shanghai, China). A 1× sample buffer was added to the samples; they were boiled for 8 min and stored at -20 °C until use.

After SDSPAGE and electrophoretic transfer, the membrane was incubated with the anti-gamma H2A.X (phospho S139) antibody (#ab228655, 1:2000 dilution) from Abcam (Cambridge, England) to detect the level of phosphorylated γH2A.X. Histone H3 was used as the internal control, which was detected using the antibody (#A2348, 1:5000 dilution) from ABclonal (Wuhan, China). After thorough washing, the membranes were probed with the HRP-conjugated goat anti-rabbit secondary antibody (#BA1050, 1:5000 dilution) from Boster (Wuhan, China). A quantitative Western blot imaging system (FluorChem Q, Alpha Innotech, San Leandro, MA, USA) was used for the photography of the blots. The protein bands were analyzed using ImageJ2 [66] for relative quantification.

### 4.9. ROS and C-Reactive Protein Assays

To investigate the effects of cold exposure and rewarming on tissue ROS levels, both the WT and 3151b mutant larvae were treated with the same conditions as those for Western blotting. The samples were homogenized in 100 μL PBS, centrifuged at 4 °C, 6000× *g* for 15 min. The supernatant was collected and subjected to ROS analysis using the ROS Assay Kit from Nanjing Jiancheng Bioengineering Research Institute (Nanjing, China). The protein content of the samples was also measured as described above. The fluorescence of the samples was normalized to protein content (fluorescence unit/mg protein). Finally, the relative ROS level of the samples was expressed as a ratio to that of the untreated WT larvae.

For C-reactive protein (CRP) assays, the larvae subjected to cold shock (exposed to 10 °C for 12 h) and recovery (exposed to 10 °C for 12 h followed by recovery at 28 °C for 6 h) were collected and homogenized in 150 μL PBS. The lysate was centrifuged at 4 °C, 6000× *g* for 15 min. The supernatant was collected and used for total protein and CRP measurements. The content of CRP (μg/g protein) was detected using the fish C-reactive protein ELISA kit from MEIMIAN (Yancheng, Jiangsu, China).

### 4.10. Statistical Analyses

Statistical analyses were performed using IBM SPSS statistics 25.0 (IBM Inc., Armonk, NY, USA). The significance of difference between the experimental groups was analyzed with *T* tests of independent samples. The survival time of the adult fish under lethal cold stress was analyzed using the Kaplan-Meier method (log rank) to test the significance of difference between the mutant and WT.

## 5. Conclusions

Tmem39b is an ER-localized transmembrane protein that plays a pivotal role in regulating the cold resilience of zebrafish. The deficiency of *teme39b* sensitized both zebrafish larvae and adults to lethal cold stress. The adults of the *tmem39b* mutant line demonstrated more severe tissue damage and cell apoptosis upon cold exposure. The results of RNA-seq analyses and DNA damage measurements suggest that zebrafish Tmem39b facilitates the establishment of cold resistance under cold exposure and the restoration of cellular homeostasis upon subsequent rewarming. The multifaceted functions of Tmem39b in regulating the cold resistance of fish could be ascribed to its effects on both cold- and rewarming-activated genetic programs. The former is mainly related to transcription regulation and chromatin regulation, while the latter is mainly associated with immune responses. The up-regulation of transcription factors, such as *fosab* and *egr1*, upon cold exposure was Tmem39b-dependent. Furthermore, the damage-induced up-regulation of the immune genes, such as *c3a.3*, *il11a*, and *mmp9*, during rewarming after cold exposure was also facilitated by Tmem39b. Tmem39b’s roles in regulating cold resistance was largely independent of endoplasmic reticulum stress response and autophagy. Moreover, the up-regulation of tissue CRPs, a class of important immune factors, during recovery from the cold-induced tissue damage was significantly attenuated in the zko3151b mutants, suggesting that zebrafish Tmem39b may regulate immune responses through mediating the CRP’s secretion. Moreover, our data shed a novel light on the biological pathways determining the cold-resistance performance of fish.

## Figures and Tables

**Figure 1 ijms-23-11442-f001:**
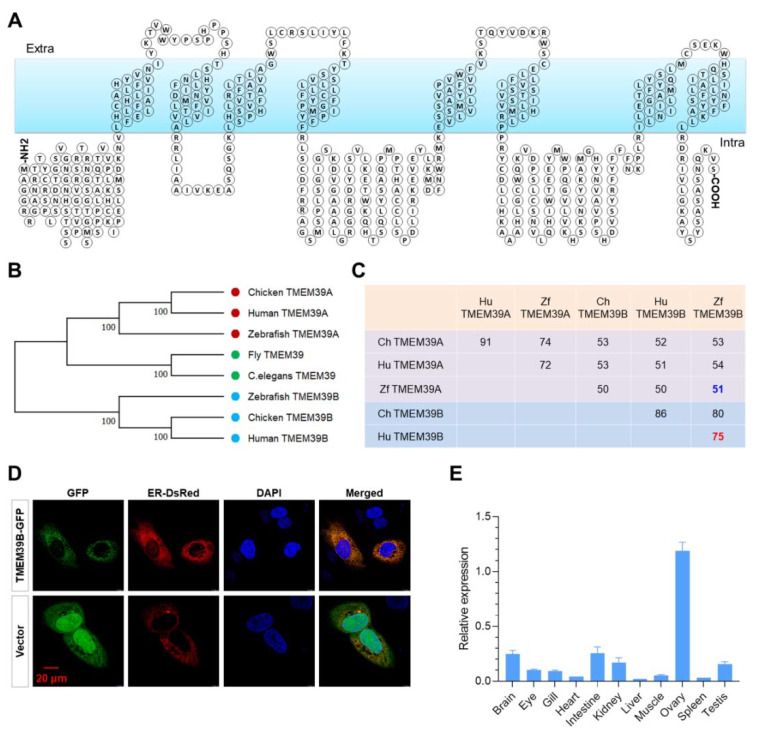
Molecular characteristics of zebrafish Tmem39b. (**A**) Predicted structure of zebrafish Tmem39b protein, which contains 8 transmembrane domains. (**B**) A phylogenetic tree of TMEM39 proteins from the representative species. (**C**) Sequence identity between the TMEM39 proteins of several vertebrate species. (**D**) GFP-tagged zebrafish Tmem39b localizes to the endoplasmic reticulum. (**E**) Distribution of *tmem39b* transcripts in the tissues of adult zebrafish.

**Figure 2 ijms-23-11442-f002:**
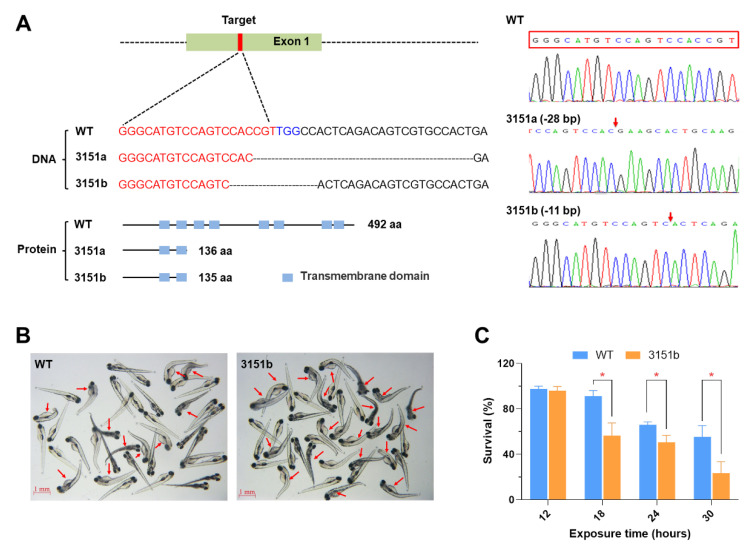
Knockout of the *tmem39b* gene sensitized zebrafish larvae to lethal cold stress. (**A**) Genotypes of the mutant lines. The mutations result in truncated proteins containing only 2 transmembrane domains. The genotypes are confirmed by DNA sequencing. The red box indicates the target sequence and the red arrows demonstrate the mutation sites. (**B**) Photos of the WT and zko3151b mutant larvae following exposure to cold-warming stress. The larvae were exposed to 10 °C for 24 h and photos were taken following the return to 28 °C for 12h. The red arrows point to dead fish. (**C**) Survival rates of WT and zko3151b mutant larvae following exposure to 10 °C for the indicated time points followed by recovery at 28 °C for 24 h. *, *p* < 0.05 (*n* = 4).

**Figure 3 ijms-23-11442-f003:**
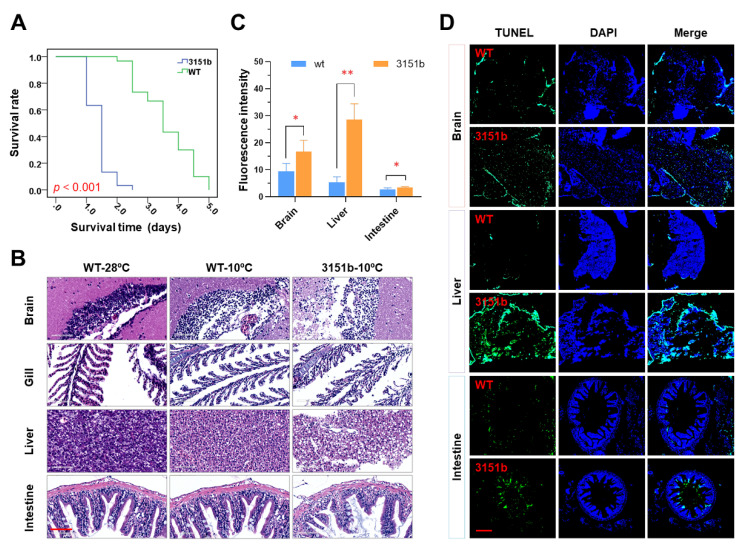
Dysfunction of the *tmem39b* gene decreases the cold resistance of adult zebrafish. (**A**) Survival rates of WT and zko3151b adults upon exposure to lethal cold stress. (**B**) Photos of the hematoxylin and eosin (H&E)-stained tissue sections. The scale bar represents 60 μm. (**C**) Fluorescence intensity of the TUNEL (terminal deoxynucleotidyl transferase-mediated dUTP nick-end-labeling)-stained tissue sections. *, *p* < 0.05; **, *p* < 0.01 (*n* = 3). (**D**) Fluorescence images showing more severe cell apoptosis in tissues of the zko3151b mutants in comparison to the WT. The scale bar represents 150 μm.

**Figure 4 ijms-23-11442-f004:**
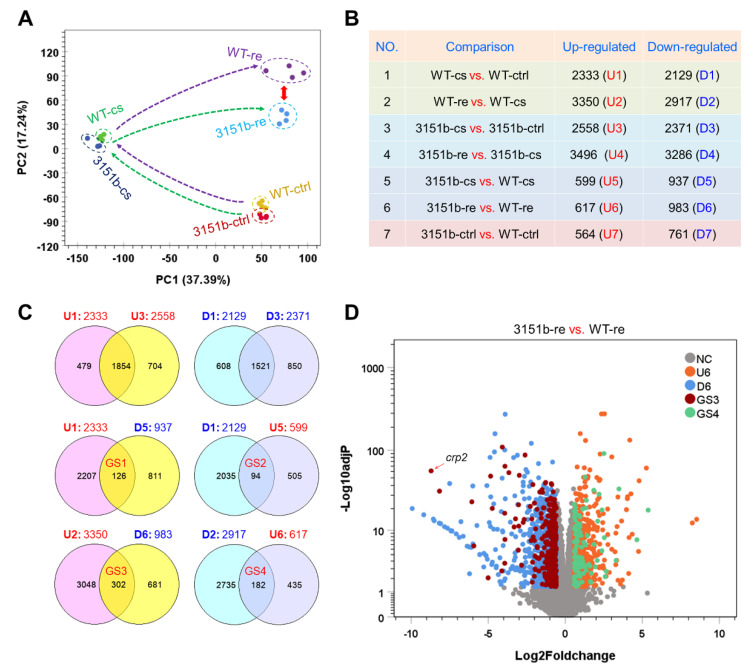
Effects of Tmem39b dysfunction on gene transcriptional expression of zebrafish larvae during cold exposure and rewarming. (**A**) Results of principal component analysis (PCA) illustrate the overall change in gene expression between different treatments and fish lines. The dashed arrows show the trajectory of gene expression alteration during cold exposure and rewarming. The red double-headed arrow demonstrates the augmented distance between the WT and zko3151b mutants during the recovery phase. (**B**) Numbers of the genes differentially expressed between different treatments and fish lines. The up- and down-regulated gene sets (U and D) are numbered sequentially according to the comparisons. (**C**) Venn diagrams demonstrate the number of genes affected by *tmem39b* mutation under different circumstances. (**D**) A volcano plot indicates the genes affected by Tmem39b deficiency during rewarming. Representative genes induced by rewarming and negatively affected by *tmem39b* mutation are shown. NC, not changed; U6 and D6, the same as in (**B**); GS3 and GS4, the same as in (**C**).

**Figure 5 ijms-23-11442-f005:**
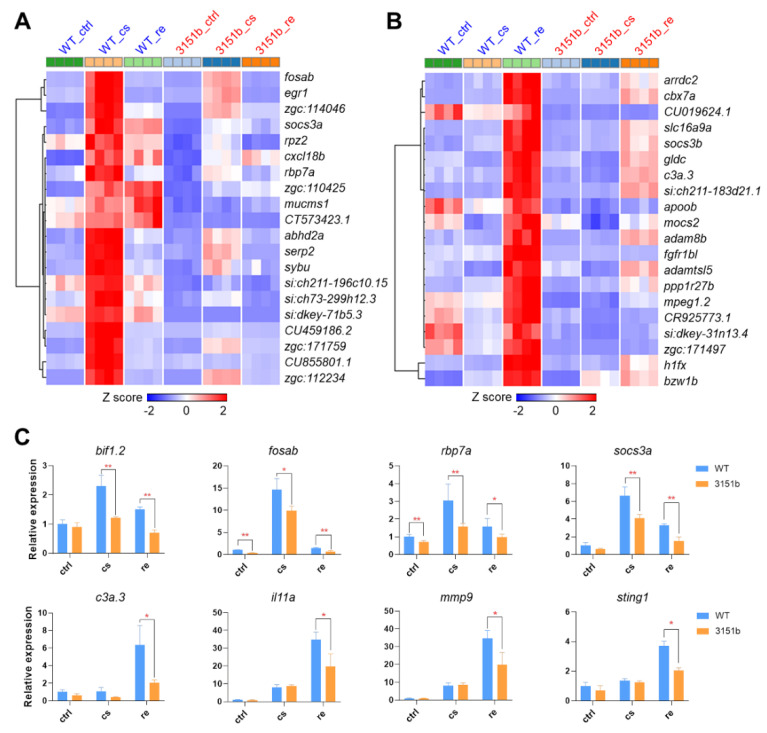
Representative genes affected by Tmem39b dysfunction upon cold exposure and rewarming. Heatmaps demonstrate expressions of the representative up-regulated genes upon cold exposure (**A**) and rewarming (**B**). The up-regulation of these genes is attenuated in the *tmem39b* mutants in comparison to the corresponding genes in WT. (**C**) Transcriptional expression of the representative genes determined using qPCR. *, *p* ≤ 0.05; **, *p* ≤ 0.01 (*n* = 4).

**Figure 6 ijms-23-11442-f006:**
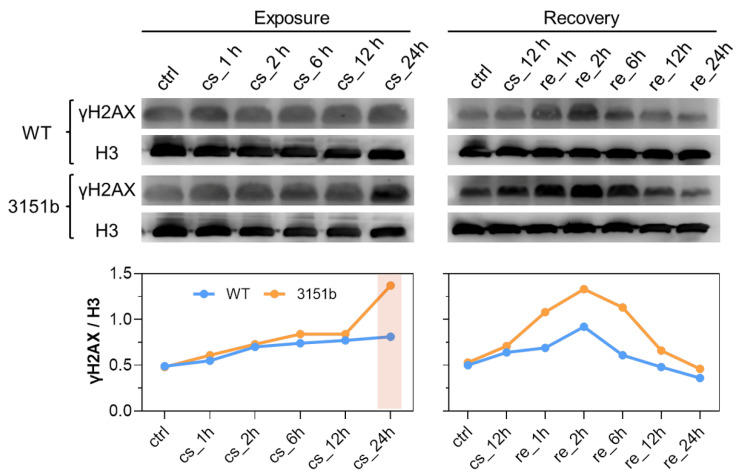
Tmem39b protects the organism from cold-warm stress-induced DNA damage. The upper panel contains Western blots for the WT and *tmem39b* mutant samples collected at the indicated time points. The membranes were immunostained with antibodies against γH2AX and histone H3. The line charts illustrate the γH2AX/H3 ratio of optical density obtained by analyzing the corresponding protein bands.

**Figure 7 ijms-23-11442-f007:**
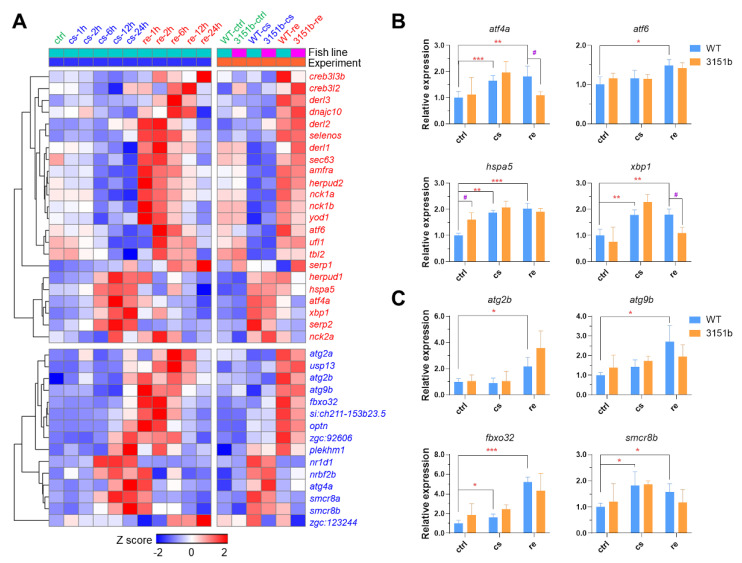
Effects of Tmem39b dysfunction on endoplasmic reticulum stress response and autophagy. (**A**) A complex heatmap indicating the transcriptional expression of genes involved in endoplasmic reticulum stress response (red) and autophagy (blue). The genes were up-regulated at certain time points upon cold exposure or recovery under a normal temperature. The left part of the heatmap indicates the results of a previous study and the right part displays the data of this study. The rows of the heatmap represent genes and the columns show the averaged expression of the genes during different conditions. The row Z scores are calculated separately for the two experiments. Expression of representative genes involved in ER stress response (**B**) and autophagy (**C**) are determined by qPCR. *, *p* ≤ 0.05; **, *p* ≤ 0.01; ***, *p* ≤ 0.001; #, *p* ≤ 0.05 (*n* = 4). “#” indicates significant difference in gene expression between WT and zko3151b under the same condition.

**Figure 8 ijms-23-11442-f008:**
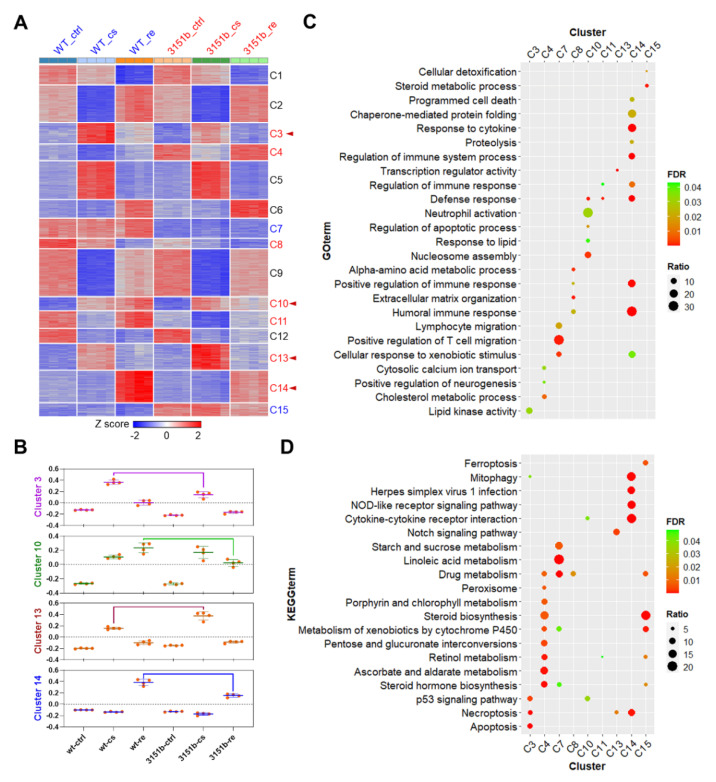
Biological processes and pathways influenced by *tmem39b* mutation. (**A**) A heatmap demonstrating the expression of DEGs across all the samples. The DEGs are classified into 15 clusters according to their abundance using the k-means clustering algorithm. The cluster names are shown on the right of the heatmap. The clusters regulated by both cold-warm stress and *tmem39b* mutation are shown in red, while those only influenced by a Tmem39b deficiency are shown in blue. The clusters associated with the functions of Tmem39b in determining cold resistance and damage repair are highlighted with red arrow heads. (**B**) Eigengenes of the gene clusters associated with the functions of Tmem39b. GO terms (**C**) and KEGG pathways (**D**) were enriched for the gene clusters affected by *tmem39b* mutation. FDR, false discovery rate; ratio, percentage of the identified genes to all the genes associated with a certain term.

**Figure 9 ijms-23-11442-f009:**
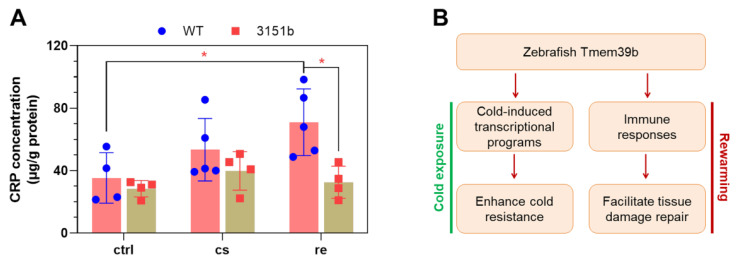
Zebrafish Tmem39b regulates the production of C-reactive protein (CRP) during rewarming and a working model for Tmem39b’s function in determining cold resilience. (**A**) Deficiency of *tmem39b* attenuates the up-regulation of CRP in zebrafish larvae during the recovery phase following cold exposure. CRP concentration is normalized as μg/g protein. All the data points are shown. *, *p* ≤ 0.05. (**B**) A working model for zebrafish Tmem39b’s function.

## Data Availability

The data presented in this study are available in the article and the Appendix A. The RNA-seq data of this study are deposited in the NCBI Sequence Read Archive (SRA) with the BioProject accession number PRJNA857857.

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
