# Peer review of "Understanding the Function and Mechanism of Zebrafish Tmem39b in Regulating Cold Resistance"

_ijms, 2022, doi:10.3390/ijms231911442_

Round 1
Reviewer 1 Report
The authors performed transcriptome sequencing and qRT-PCR to characterize gene transcription and phenotypic changes in zebrafish with knockout of the tmem39b gene. In general, the overall writing is good. Numerous data were generated to support the main conclusions. However, minor revisions are required before acceptance for publication.
Abstract:
1. Provide the full name for any abbreviated term at its first appearance in the main text (such as TMEM39B in lines 20-21). It is very strange that tmem39b) is the core gene of this project, but we couldn’t find its full name throughout this manuscript.
Results:
2. Lines 136 & 140: Tmem39b should be italic. Please pay attention this issue that gene names should be italic throughout this manuscript.
3. Lines 269-272 and others: Change “expression” to “transcription” when necessary, since in most cases only transcriptome sequencing and qRT-PCR was performed for gene quantification.
4. Provide an Abbreviation list for those abbreviated genes. There are so many abbreviations without details throughout the manuscript.
Discussion:
5. The authors are recommended to prepare a figure to summarize the detailed mechanism of tmem39b ‘s cold resistance.
Conclusions:
6. Lines 692-700: This paragraph should be expanded with more results.
Data availability:
7. Lines 725-726: Transcriptome sequencing reads and related details should be deposited at NCBI for public availability and comparisons.
Reviewer 2 Report
In the manuscript (MS) titled “Understanding the function and mechanism of zebrafish Tmem39b in regulating cold resistance through phenotypic and transcriptomic analyses”, the authors knocked out the tmem39b gene in zebrafish and checked its regulation mechanism. Overall, the MS is well designed and well written with enough data, I do not have major concerns on this MS, but only some minor comments that I hope the authors can address to improve the MS.
1. The title is too long, it should be shorter and clearer to make it easily read through.
2. Line 82, “However, prolonged ER stress response can activate apoptotic 82 signals [42] and were transcriptomic analyses”, were is not correct.
3. Line 113, a “,” should after “Zebrafish Tmem39b consists of 8 transmembrane domains:.
4. In Fig.1D, make sure the scale bar is correct; What does vector stand for? Why the GFP in TEEM39B-GFP is so weaker than that in vector? In “GFP-tagged Zebrafish Tmem39b”, the Zebrafish should be zebrafish.
5. Line 143, “Accession number of the mutant lines are zko3151a (3151a, -28 bp) and 144 zko3151b (3151b, -11 bp)”, are should be is.
6. Line 202, “effects of Tmem39b mutation on gene expression was”, was should be were.
